# Lived Experience after Bariatric Surgery among Patients with Morbid Obesity in East Coast Peninsular Malaysia: A Qualitative Study

**DOI:** 10.3390/ijerph19106009

**Published:** 2022-05-15

**Authors:** Siti Faezah Gullaam Rasul, Nani Draman, Rosediani Muhamad, Zainab Mat Yudin, Razlina Abdul Rahman, Samsul Draman, Mohd Nizam Md Hashim

**Affiliations:** 1Department of Family Medicine, School of Medical Science, Universiti Sains Malaysia, Kota Bharu 16150, Malaysia; siti_faezah87@student.usm.my (S.F.G.R.); rosesyam@usm.my (R.M.); razlina@usm.my (R.A.R.); 2School of Dental Science, Universiti Sains Malaysia, Kota Bharu 16150, Malaysia; drzainab@usm.my; 3Department of Family Medicine, Kulliyyah of Medicine, International Islamic University Malaysia, Jalan Sultan Ahmad Shah, Kuantan 25200, Malaysia; nurin@iium.edu.my; 4Department of Surgery, School of Medical Science, Universiti Sains Malaysia, Kota Bharu 16150, Malaysia; drnizamkk@usm.my

**Keywords:** lived experience, bariatric surgery, obesity, morbid obesity

## Abstract

Living with morbid obesity is challenging since it affects various dimensions of life. Sustainable weight loss via bariatric surgery helps people suffering from morbid obesity to lead a healthy and meaningful life. This study highlights the challenges before bariatric surgery and the impact on one’s life. A phenomenological approach was employed using in-depth interviews with 21 participants (15 females and 6 males) who had undergone surgery at least 6 months prior to the study with the mean age of 42.6 years. Due to excess body weight, the physical limitation had a serious negative impact on their social life, making them targets of bias and stigmatisation. Surgery was the best option for them to attain sustainable weight loss and to lead a new life. However, a few participants struggled with the side effects of surgery. Five themes were discovered, namely, (1) social restraint; (2) experiencing bias and stigmatisation; (3) bringing new life; (4) boosting self-esteem; and (5) facing the negative side of surgery. This study serves as a platform to explore the difficulties faced by people with morbid obesity and the changes that the participants experienced after the surgery for future intervention to curb the rising number of people with morbid obesity.

## 1. Introduction

Obesity has become a global threat that affects people from around the world. Worldwide, over six hundred fifty million adults, amounting to 11% of men and 15% of women, live with obesity [1]. Likewise, many Asian countries face this rising obesity trend as they undergo socioeconomic and lifestyle transitions as a result of rapid globalisation and urbanisation. Unfortunately, with the rise in urbanisation, individual physical activity has decreased, contributing to upper body adipose tissue deposit and substantial body mass index (BMI) rise [2].

The growing number of people living with morbid obesity has led to serious health consequences and caused significant obesity-related health care expenditure [3]. Aside from poor physical health, an individual living with obesity also suffers from poor mental health due to weight stigma [4]. They face perceived weight discrimination in various angles of life such as employment, healthcare, and interpersonal relationships [5]. Therefore, this surgery stands as a tool to solve one’s weight problems and as a weapon to combat all comorbidities, including physical and biopsychosocial problems attributed to excessive weight. In the midst of overcoming this muddle, bariatric surgery stands not only as strongly established but also as one of the most effective and safe treatments for severe obesity [6]. Bariatric surgery, including gastric bypass, sleeve gastrectomy (SG), and laparoscopic gastric banding, often helps in reducing excess weight loss within one year of surgery for a prolonged duration [7]. In addition to the substantial and sustained effects on weight, this surgery significantly ameliorates obesity-attributable comorbidities such as type 2 diabetes mellitus, cardiovascular disease, hypertension, dyslipidaemia, and sleep apnoea in most patients [7]. 

According to the International Federation for the Surgery of Obesity and Metabolic disorders (IFSO) Worldwide Survey 2014, about 634,897 bariatric operations were performed worldwide in 2016 [8]. Various surgical options are now available, and they are continuously evolving globally. 

In Malaysia, the first bariatric surgery, an open vertical banded gastroplasty, was performed in 1996, followed by a laparoscopic gastric bypass in 2001, laparoscopic adjustable gastric banding in 2005, and laparoscopic sleeve gastrectomy (LSG) in 2006 [9]. There has been a steady increase in the number of bariatric cases performed annually in Malaysia. The constant rise in popularity of bariatric surgery in Malaysia can be largely attributed to inclination in awareness among healthcare providers regarding the availability of surgical management of obesity both in the private and government sectors, and obesity-related comorbidities, rigorous public awareness campaigns by the Ministry of Health, and the proactive involvement of both print and electronic media in providing information on the deteriorating effect of obesity. 

Various studies were conducted globally to explore the changes that the participants’ experienced in their life after the surgery. Previous studies describe the positive consequences of this surgery in many domains of life, including physical health, sexual life, and recovery of self-esteem [10,11,12]. However, such studies are still lacking among the Asian population, as many studies conducted were focused on specific disease resolution and cardiovascular risk reduction [13,14]. Data on the biopsychosocial aspect of life, spiritual health, and self-esteem are still unexplored among Asian populations with different cultures and ethnicities. 

This research adopted the humanistic theory using Maslow’s hierarchy of needs to explore the changes that the participant experienced after going through bariatric surgery. Abraham Harold Maslow was an American psychologist who first introduced his hierarchy of needs in a 1943 paper entitled *A Theory of Human Motivation* [15]. This five-stage modal includes physiological needs, safety needs, love and belongingness needs, esteem needs, and self-actualisation needs. Over several decades, Maslow refined his theory and stressed that the order of needs might be flexible based on external circumstances or individual differences and not necessarily follow the hierarchy orders. This framework helps us understand that every person is capable and desires to move up the hierarchy toward a level of self-actualisation when these growth needs have been reasonably satisfied. 

Understanding participants’ experiences and views about this life-saving surgery is essential for healthcare workers to reach more patients in need of this intervention. This study aims to explore the challenges faced by patients before the surgery and the impact of the surgery on patients’ quality of life. 

## 2. Materials and Methods

This qualitative study was conducted with a phenomenological approach by exploring post-bariatric patients’ stories, empowering them to share their very personal life experiences and challenges before and after the surgery. This phenomenological approach sets aside biases and preconceived assumptions about feelings, human experiences, and responses to a certain scenario. This enables us to delve into the viewpoints and perceptions, as well as the feelings and understanding of those who have been in this position [16]. This research only involved post-bariatric surgery patients proficient in Malay and English and this was approved by the Human Research Ethics Committee of USM (USM/JEPeM/19100567).

### 2.1. Setting

This research was conducted at Hospital Universiti Sains Malaysia (Hospital USM) in Kelantan, a state located in Northeast Malaysia, with Malay Muslim accounting for 94% of the population [17]. The hospital also receives referrals from other tertiary hospitals on the east coast of Malaysia.

### 2.2. Participant

Participants who met the inclusion criteria, those who had the surgery at least six months ago, were contacted by sharing posters and messages via a short messaging application (WhatsApp) to the specific WhatsApp group of bariatric patients who had undergone the surgery. Participants aged less than 18 years old were excluded from this study. Participants were recruited from those who contacted the researcher to participate in this study based on the details shared via WhatsApp. The researcher arranged an appointment for an interview session with the patients. Snowballing technique was also used to increase the number of participants recruited. Recruitment of patients was stopped once recurrent patterns had emerged during the interview, and theoretical saturation was achieved from 21 interviews [18].

### 2.3. Procedure 

Data were collected through face-to-face, in-depth interviews using a semi-structured questionnaire consisting of several key questions to assist the researcher in defining the areas to be explored in more detail. The interviews were arranged at a time and place convenient to the participants. These were also conducted by the same researcher, a family medicine trainee who had undergone training to conduct a qualitative study. Most of the interviews were conducted in a private room at the Department of Family Medicine, School of Medical Science Health campus USM, while three interviews were conducted at the patients’ office. All the researchers have no prior relationship with the patients. Consent for an audio recording of their interviews was obtained before conducting the interview. Participants were then requested to complete a sociodemographic questionnaire. Interviews were conducted in English and Malay, depending on the patient’s preference. The session was started by asking open-ended questions such as: ‘What do you know about obesity?’, ‘How does obesity affect your life?’, and ‘What is the impact of bariatric surgery on your life?’ before exploring the details about patients’ experiences. This questionnaire was piloted on three patients before the real interview to evaluate the questionnaire’s appropriateness, acceptance, and validity. The total interview time per patient ranged from 60 to 90 min. All interviews were audiotaped, kept secure, and transcribed verbatim into text before being analysed. 

### 2.4. Data Analysis

The transcriptions were coded using the NVivo 12 (Qualitative Research Computer Analysis Package) software. Relevant themes were extracted from the transcribed data by using the inductive method according to the thematic analysis approach of Braun and Clarke [19]. In the beginning, all the researchers (S.F.G.R., R.M., N.D., Z.M.Y. and R.A.R.) thoroughly reviewed the first five transcripts to familiarise themselves with the patients’ general views and perspectives. Later, the main researcher created an initial list of codes in NVivo^®^ (Microsoft Corporation, Redmond, WA, USA), and transcripts were coded precisely. A rereading of each participant’s transcript was performed until all the researchers made sense of each word and sentence, discovered repeated patterns of meaning, and later made sense of the participant’s stories as a group. Each transcript was examined in detail before collecting the identified themes and organising them into an interconnected framework (themes, subthemes, and axial coding). To ensure proper coding, validity, and reliability, the research supervisors (N.D. and Z.M.Y.) collaborated with the main researcher to recheck the transcript’s codes. A co-researcher (R.A.R.) with a background in obesity further examined and discussed the coding and themes for all transcripts to generate helpful ideas. All conceptual differences in the thematic analysis were addressed and revised. Subsequently, other preliminary themes were developed. Finally, a mutual consensus was reached among the researchers on the final codes, including all the themes, subthemes, and axial coding. The transcribed material was sent to the participants to verify the content of the results. None of the participants had a contrary view.

## 3. Results

### 3.1. Characteristics of the Participants

This study involved 21 adult participants (15 females and 6 males). All of them were Malay Muslim with a mean age of 42.6 years. Most of them were married and working in the education sector. Twenty of them had comorbidities before the surgery. The mean pre-operation BMI was 51.4 kg/m^2^ and the mean post-operation BMI was 38.6 kg/m^2^. 

Table 1 summarises the detailed characteristic of the participants.

### 3.2. Themes

Five main themes were identified, namely: (1) social restraint, (2) experiencing bias and stigmatisation before the surgery, (3) bringing new life, (4) boosting self-esteem, and (5) facing negative side of surgery (Table 2).

### 3.3. Theme 1: Social Restraint

Daily life activities were challenging for all the patients when they were once living with morbid obesity. Their excessive weight had so many negative influences in multiple domains of their life, and the health-related issues were the most striking of all. Excessive weight has a negative impact on human biomechanics. Multiple joint pain and poor physical fitness had made their daily life difficult. Moreover, the heavy weight made them easily tired and breathless due to high energy expenditure. These have led many people with morbid obesity to restrain themselves from social life. For example, one male participant, P5, remarked:


*“For everything that we do, all our movements are restricted. Before the surgery, I will drive my wife to the shopping complex; I’m honestly tired. I will just let her get into the mall alone to do the shopping. I will wait in the car because I’m easily tired.”*


The extreme weight also brought a negative impact on their spiritual life. For Muslims, physical health is essential for executing a spiritual act, which entails standing, bending down, and prostration. Men are urged to perform the five daily prayers at a mosque, particularly the Friday prayer, which must be performed in the mosque. Many men with obesity feel reluctant to go to the mosque as walking to the mosque will leave them breathless with unpleasant body odour due to excessive sweating. P16 shared his experience, *“I will leave to the mosque early on every Friday to get the nearest parking spot as possible… so I don’t have to walk far away to reach the mosque because… I will sweat a lot and that makes me uncomfortable.”* Moreover, their restricted limb movements made it difficult for them to perform a spiritual act with perfection. For example, P16 further said that his big physical appearance affects his ability to perform the prayer.


*“Taraweeh during Ramadhan, Friday prayers… I will always feel embarrassed because I will take a lot of space in the prayer line, and Tahayat akhir will always be a problem because I have to bend my legs. […] It was quite problematic. Because I might push the person next to me when I tried to bend my leg, but yeah, it was uncomfortable for me.”*


Even at the workplace, many participants struggle to carry out tasks assigned to them. Working in the education sector requires most participants to monitor the students, and social connection with students is critical to maintain a strong bond with them. Unfortunately, being obese and physically unfit had kept them tired and less interactive with their students. P21, a teacher, expressed her difficulties.


*“I can’t start teaching for the first 10–15 min because I will be breathless, gasping for air, especially if I had to attend classes at higher levels. I have to rest for about 10 min before starting the class, and when I sweat… I feel uncomfortable to walk beside my students to check on their work.”*


Aside from the physical limitation, the participants were socially restrained due to a lack of attractive attire options. There are a lot of clothing designs for average-sized people, but there are only a few options for obese individuals, making them believe that fashion is not meant for them. P8 said,


*“The saddest part was the clothing. Other people look smart. When I was at my maximum weight, about 140 kg, I had to wear size 4XL. Everything I wore was tight. It was uncomfortable for me. I prefer to be at home. […] I had to buy my clothes at a bundle shop. If I go to the mall, the clothes were very expensive, and there weren’t many choices.”*


Furthermore, inadequate public facilities have caused many people who are morbidly obese to refuse to travel. Public transportation, particularly aeroplanes, are frequently small. From the stairs to the washrooms, the aeroplanes are all catered for people with regular weight. P8 further shared his experience:


*“Previously, when I had to travel by flight, I had to have extra belts, which means they had to join two belts for me; I can’t go for Haj. I can’t be sitting on the flight like that. Can you imagine, the belts were tight, and the space was small. I can’t even unfold the table in front of me. I was suffering.”*


Social restraint among people with obesity is also attributed to embracing life events they face. Their large stature made them walk with an unsteady gait, making them prone to fall. Moreover, their heavy weight also made them vulnerable to falling from a broken chair, since most public places in Malaysia provide people with plastic chairs that cannot handle heavy weight. P10 stated:


*“Sometimes I became sad thinking that I need to pray, especially when travelling. Because there won’t always be chair prepared. Even if there were chairs available…those were plastic chairs with the mosaics the chair legs tent to spread. I’ve fallen about two to three times. I felt so embarrassed with the people around me. I fell on my buttocks while praying, sitting on a plastic chair. Because the plastic chair can’t support my heavy weight. The chair legs started to spread and break.”*


### 3.4. Theme 2: Experiencing Bias and Stigmatisation

People who are overweight and obese are always vulnerable to discrimination and body shaming. The labelling thrown to them, even worse coming from their loved ones, had caused psychological stress among them. P10 shared her bitter experience when she was once a victim of body shaming from her children:


*“My children always say that ‘Mommy, you walk like a penguin’ because I can’t walk… it was till that extend …Limited. It was all limited.”*


Facing people in daily life had become torture. The discriminating gesture by the public due to their heavy weight was indeed soul breaking for many of the participants. They were often left feeling embarrassed in public places when people acknowledged them for their huge physical appearance. P18 had expressed feeling stressed due to the discriminating gesture by the public toward her.


*“The worst was mental torture. Because people looked at me like a monster walking. I felt sad. Some people pulled people next to them as I walked to look at me. There was once a person who burst into laughter looking at me. Truly I felt myself like a monster walking… really mental torture.”*


At times, what kept them timid was not about the surrounding people, but the self-perceived body image that they look imperfect. Their thought of being imperfect has taken away their self-confidence. Such internalised stigmatisation could help explain the low self-esteem and poor body image among adults with obesity, which led them to self-stigmatisation. P16 expressed concern about being judged by his students.


*“Because I teach, I’m a lecturer. I feel low self-esteem with my students and whenever I go into the lecture. I will set up my computer connected to the PowerPoint projector; I will sit and talk and press the button for the slides and wouldn’t even round in the classroom. Because I’m so scared and embarrassed about being judged on my big tummy back then.”*


Even at the workplace, many participants experience discrimination. Judgemental comments on their work quality left them unappreciated. Their efforts were unrecognised, parallel to the popular belief that people with obesity are lazy. P13, stressed:


*“Before the surgery, my weight was about more than 100 kg. […] My supervisor frequently remarks on how slow I work. Only a small amount of work was completed. I sit more compared to working. “I’m too lazy to attend to the customer.”*


### 3.5. Theme 3: Bringing New Life

The struggles and humiliation they faced became strong reasons for them to opt for surgery. The determination to end their misery was stronger than their apprehension of undergoing a major surgery. Once they met the doctor at the surgical outpatient department, all participants agreed to surgery due to their strong willpower and family support. Indeed, for almost all the patients the surgery was a great turning point. They gained significant advantages in various scopes of life, including physical health, spiritual ritual capacity, and sexual life. It also gave them confidence to venture into new hobbies and activities. They believe that the surgery has given them a second chance to live and taste the beautiful life they always dreamed of.

#### 3.5.1. Improved Physical Health

Most of the patients suffer from multiple diseases due to morbid obesity. Diabetes, hypertension, hyperlipidaemia, obstructive sleep apnoea (OSA), osteoarthritis, and other diseases are not uncommon among people with morbid obesity. P21, a lecturer, was diagnosed with diabetes, hypertension, and hyperlipidaemia about three years before the surgery. Her dietary habits made her blood sugar level uncontrolled. The busy schedule kept her physically inactive, making it difficult to control her high blood pressure, and rising cholesterol levels made her more vulnerable to cardiovascular complications. She experienced great improvement and recovery from the diseases she suffered for years following the surgery. P21 expressed that she was glad that she had the surgery, which brought tremendous health benefits.


*“…felt as if I was given a new life. Diabetes, high blood pressure, high cholesterol, are all gone.”*


P16, a lecturer, suffered from obstructive sleep apnoea. He had a severe snoring habit. His snoring was so loud that it could be heard outside his house. His mother once said that he snores like a motorboat. At times, he also wakes up at night, as he felt the sensation of choking due to his excessive neck fat. Poor quality of sleep made him tired and less productive. With the help of surgery and adopting a healthy lifestyle, he recovered from OSA.


*“So after treatments of bariatric surgery, my OSA reduces. I have no more snoring issues anymore. Just after three months of surgery.”*


P16 further added that he had pigmentations at his armpit and neck due to insulin resistance when battling morbid obesity. He even suffered from severe acne conditions. He sought medical advice multiple times from many skin specialists, yet his cystic acne did not resolve. He also spent a lot of money buying expensive supplements and skincare products to help him attain healthy skin. Unfortunately, not much improvement was seen. About five months after the surgery, he noticed the resolution of the acne and the disappearance of all the pigmentation that he had.


*“No cystic acne anymore. Back then, it was huge cystic acne. […] About five to six months after the surgery, all the pigmentation at the armpit and neck cleared out. My armpit looks brighter now. My neck as well.”*


Many of the participants also suffered from osteoarthritis. The incapacitating knee pain caused many detrimental consequences in their daily lives. Standing or walking for a long time exacerbated the pain, and for women activities such as cooking and cleaning the house were challenging. P18, a teacher, shared that she was diagnosed with bilateral knee osteoarthritis at 40 years old. She was suffering from debilitating knee pain at home and work, making it difficult for her to perform her tasks. It was difficult for her to climb up the staircase to the second or third floor to attend her classes. Moreover, standing for long hours conducting the class often exacerbated her knee pain. The pain was so excruciating that she needed to be on an oral pain killer frequently, and at times she required intra-articular injections to alleviate the pain. After the surgery, the knee pain reduced significantly, and she barely required a pain killer.


*“Now I still have knee pain. Knee pain is still one of my problems […] but better now than before surgery.”*


#### 3.5.2. Regaining the Ability to Perform Religious Rituals

With the extreme weight loss through the help of surgery, many participants had improvement in their freedom to perform spiritual acts. Performing prayers was more pleasant without struggling to lift an extremely heavy body weight. The surgery lifted the physical limitation, making them capable of performing all the spiritual acts they wanted. Reciting Al-Quran, which requires one to manipulate their breathing, has become easier. They need not struggle or feel breathless to complete one surah in the Quran. P3 shared that she could perform her prayer longer and complete reciting the surah that she preferred.


*“Now ok. If I want to perform Salatul Hajat. I can recite Ayatul Kursi seven times…I can stand longer. Now everything is ok.”*


Similarly, P1 values her spiritual act more as she feels more connected to her religion. After her weight reduction surgery, she can now perform her spiritual acts unhindered.


*“I always get up early to perform Salatul Tahajjud because I feel the movement has been lighter. There are so many things we can do now. Because we feel relaxed. That’s precious for me…because I can do sujood. It feels different. We can perform prayers as often while sitting on a chair, and it’s easy. But once we do it properly, the sujood the rukuk, we can feel the difference. That’s the best feeling for me.”*


#### 3.5.3. Pleasant Intimacy

Obesity affects sexual health. Many factors related to obesity contribute to unpleasant sexual intimacy. Their unattractive look, low self-esteem, and physical limitations with multiple comorbidities stand as barriers to healthy sexual life. P21 felt it was such misery to have sexual intimacy due to her heavy weight restricting her physical movement.


*“Previously, I felt so uncomfortable. It was difficult for me even to lift my leg. Truly for people, sexual intimacy is something enjoyable, but for me, it’s a torture doctor. I don’t like it. I don’t like it. I don’t enjoy doing it.”*


Moreover, the participants’ libido was negatively affected by what they perceived as an undesirable physical appearance due to their excessive weight. P18 expressed that for her having an intimate relationship was troublesome: *“I felt embarrassed because the fat had covered all the private part; My husband had to struggle a little, which made me unexcited about the intimacy because it became troublesome.”*

After the weight loss surgery, many of them noted improvement in their sexual life. Some participants noticed increased libido with weight loss. P10 expressed,


*“I think I had increased libido…umm…because previously I was pushing my husband away while we were together…because I was gasping for air…now that I can move…I felt that I’d increased libido.”*


Similarly, P5 experienced a positive change in his sexual life. *“After the surgery, I can have sex longer because I don’t have leg cramps anymore.”*

Many are more satisfied with their new sexual life. P20 expressed: *“…in terms of intimacy, I felt…my husband and I are more active now (laugh). My figure had become small, so I think he felt more satisfied when hugging now because I’ve become smaller.”*

### 3.6. Theme 4: Boosting Self-Esteem

Human success is built on a foundation of good self-esteem. Life would be bleak without it.

#### 3.6.1. Being Accepted

After the surgery, all the participants made a socially valued transition. They feel more accepted and valued by their family and friends. The stigmatisation that they once faced had vanished. P16 stated he feels relieved with his current life.


*“Now, I feel slightly better. I’m more outgoing. Whenever my friends invite me to join them, I’ll join them […] I think I feel people are more receptive of me. I feel more accepted now.”*


The discriminating gestures are now no more. They feel at ease being in a public place. P10 stated: “*When I get into an elevator, I sometimes use the elevator…people will look up and down at me. Now, I feel comfortable getting into an elevator.”*

P3 also experienced positive changes in her relationship with her husband. Before the surgery, she was always left at home by her husband because he was reluctant to bring her along to any family gathering or occasion. He was embarrassed by his wife’s physical appearance. He even teased her that the chairs at the function might be broken if she attends. However, he had changed after her weight loss. He was excited to bring her out.


*“Now that he sees my new appearance, he wants to bring me along wherever he goes out. If there is a family day, he will invite me to join him. It wasn’t like this before. Maybe he was embarrassed to show his wife to the people around. Now, everything is ok.”*


#### 3.6.2. Confident with New Outlook

Change in one’s physical outlook due to weight loss after the surgery had so much positive impact on the participants’ self-esteem, making them more confident. P1 stated she felt more positive about her life.


*“Now I am happy about going to work…excited about going back to my hometown…visiting my relatives…choosing my clothes. I felt the lost excitement had returned.”*


Similarly, P21 felt more comfortable attending family gatherings and functions. She no longer faces unpleasant comments regarding her physical appearance. She could fit in her desired dress. The new transformation of her physical appearance gave her the chance to be part of the community without being judged.


*“In terms of attending functions, I feel more excited because I feel confident now.”*


Being able to shop with family and friends at the mall was something many of them dreamed about, and almost all the participants enjoy shopping at a mall and even online after this weight loss surgery. Choosing nice, trendy, and fashionable clothes is now fun, especially when attending special occasions. The opportunity to choose any colour they fancy without attracting attention because of their size adds to the exhilaration. They do not need a tailor for custom-sized clothing because they can now wear pre-made outfits just like others. P21 added, *”I’m excited about attending functions. I can wear nice clothes. Back then, I couldn’t wear Kebaya; now I can wear Kebaya.”*

#### 3.6.3. Self-Betterment at Work

The benefits they attained via the surgery are also reflected in their productivity at the workplace. The disease resolutions and recharged self-esteem facilitate their better performance. New tasks have been assigned to them. Their work quality improved. They became enthusiastic about handling new projects and confident when dealing with their colleagues. P16 loves the changes he experienced with his work quality after the surgery.


*“Now, I feel better. I feel like interacting more with students, I am more positive, and I think my teaching quality is better now.”*


Similarly, P9 can now carry out her presentation well with her colleagues. She does not worry about knee pain after prolonged standing or being breathless with a long speech. Her mind is clear of her intrusive thoughts about other people’s reactions and opinions regarding her massive, unattractive physical appearance: *“I can do good a presentation; I feel I can stand in front of many people.”*

P1 also shared that she had become more enthusiastic about attending more courses, expanding her knowledge on her work scope. Her lighter body weight had made it easy for her to travel even to a different state.


*“Back then, at my workplace, I will only join simple courses that people won’t be excited to go to… like Iqra’ course. Maybe two or three people will go… now… but now… I want to join courses done outside the state. Even if courses are done in KL, I want to go. I want to be the secretariat of any programme conducted. I want to show what I can do.”*


#### 3.6.4. Exploring New Hobbies and Fun Activities

Travelling and trying new fun activities were things that many participants were reluctant to be part of because of their physical limitations and health-related issues. They were mostly worried about falls while travelling due to their heavy weight. Since the surgery, travelling has become fun and travelling alone is no longer an issue. P14 had shared: *“In January I went travelling alone so that part something I wouldn’t do it being alone. Because I was worried that it would be troublesome if anything happened, but because I was skinnier fitter, I decided, ok, I think I can do it. So, I just went to Japan alone, yeah, travelling.”*

P20 manages to bring her children for a long drive for a holiday trip. She does not feel suffocated anymore being in the car. As she became thinner, her tummy did not get in between her and the car’s steering wheel. She can drive more comfortably: *“Before this, I just bring my kids to Terengganu. Now, I can bring them to the Cameron Highland even if my husband can’t join us. A few days back, I drove to Johor using the highway. I can drive.”*

Others had started exploring new fun hobbies. P5 mentions, *“I’ve started swimming and playing football. I can do activities now that are much different from before the surgery.”*

P19 was terrified of going to the beach when she was morbidly obese. She was afraid that her legs could be trapped in the sand, causing her to get off-balance and fall. Now, she is delighted to enjoy her beach walk after the surgery: *“Now, doctor, every evening I want to go to the beach; I can play by the beachside and go to the beachside.”*

Being obese limits one from enjoying their life. They cannot enjoy most of the activities at any theme park because of their extremely hefty weight. Only after the weight loss post-surgery, many patients were eager to explore new adventures and activities. Getting into a roller-coaster was something exciting that P14 explored after surgery: *“ride roller coaster oh yeah. It’s just something that I wouldn’t do, but I did it. I mean, I did it after the surgery.”*

### 3.7. Theme 5: Facing the Negative Side of Surgery

Every surgical procedure comes with its complications. However, almost all the participants had no regrets of their decision to go through this surgery.

#### 3.7.1. Facing the Side Effects

None of the participants had life-threatening side effects post-surgery. Few of them occasionally suffer from gastroesophageal reflux disease (GERD), especially when they take spicy food or take more food than their stomach can tolerate. P16 said: *“I still face it now. If I eat any hot food, extra chilly or slightly spicy tomyam, I will always have a burning sensation on the chest.”*

Similarly, P18 had expressed: *“It’s like reflux, burning sensation in the chest. Maybe the acid vapour is bubbling to the oesophagus.”*

Three participants had experienced hair loss after the surgery. P10 mentioned, *“I have bad hair loss. I feel I’m getting bald.”* P12 had also expressed similar concerns: *“Hair loss. Now I’m just left with only a little hair. Lots of hair loss.”*

Those side effects left no regret in them as they treasure the life they are living now. P3 treasures her life like a diamond: *“Like a diamond. Care for it like a diamond. Because I just love it. I mean, we can’t leave the diamond just like that. We have to wipe and keep it shining. This means we have to be cautious with our diet and maintain the effect.”*

P10 feels that the surgery has given her a new life despite suffering hair loss: *“The surgery had given me a second chance to live. Truly without the surgery, I could have been bedridden by now. Because my bodyweight kept on increasing.”*

Indeed, the surgery was a life-changing experience, and a turning point for a better life, and they had no regret about it. P6 said: *“I’m glad that I had the surgery. I’ve never regretted it. Whenever people ask my opinion about the surgery, I support them to go for it, especially those with multiple health problems.”*

#### 3.7.2. Regaining Bodyweight

In the beginning, all the participants experienced constant weight reduction after the surgery. It was mainly because they could only tolerate fluid for about 2 weeks after the surgery. It was even longer for some of the patients; they consumed only milk for about 6 weeks until their stomachs could tolerate soft diets. About one-year post-surgery, most patients noticed that their weight was not reducing any further. Some even noticed that they were gaining weight two years after the surgery since they could now consume normal food like before. They knew that failure to be mindful of the diet had stopped further weight loss. In our studies, three patients had experienced weight gain. P18, whose weight was 178.5 kg before the surgery in 2018, expressed her regrets*: “Sadly, I’ve gained weight. I’m now 140 kg. I managed to reduce my weight to 128 kg previously.”* Similarly, P4, who was previously 145 kg before her surgery in 2019, shared her experience: *“Maybe I didn’t take care of myself well. Maybe it’s the choice of food. I realise now when I started taking sweet drinks, desserts lately …I realise that I’m gaining weight.”*

## 4. Discussion

This study underlines the challenges faced by people with morbid obesity and the impact of bariatric surgery on their life. Obesity brings so much negativity into a person’s life. It affects one’s health, mind, and soul. Many individuals can be socially withdrawn. Their huge figure severely impacts their human biomechanics, restricting their movement. The participants in our study expressed that their excessive weight stands as a huge barrier for them to carry out their social life. Their physical limitations, unfriendly public facilities, lack of attractive attire, and embarrassing life events in public places kept them timid. Life was hard for them, as they struggled each day with heavy body weight in carrying out tasks at the workplace, in performing prayers, and in fitting in the clothes they did not desire. Moreover, even if they had to go out, the public facilities surrounding them were not user friendly for people with obesity. In addition, their heavy weight also caused increased energy expenditure, leaving them breathless and constantly tired, further leading to their social dysfunction. This response reinforced the finding that the participants in this study, like many other people suffering from morbid obesity in other parts of the world, experience social restrictions in their daily lives [20]. Their heavy weight had compromised their freedom to lead a normal life, which was among the main reasons why most participants go through this surgery without a second thought.

We also found similarities with other studies regarding weight-related stigmatisation and bias experienced by participants at various angles of life. This includes stigmatisation in the workplace, as people with obesity are stereotyped as lazy [5]. They also suffer from body shaming. As for the participants in this study, the labelling is made by mostly comparing them with an animal. The labelling and body shaming were indeed soul-crushing moments, especially when it came from their loved ones. To make matters worse, the offensive and discriminating public behaviour caused them to devalue themselves [21]. These traumatising incidents further add to their low self-esteem [22].

The conventional way of losing weight via exercise and dieting did not work for most of the patients with morbid obesity. Even if they did, weight loss through these methods was transient. Bariatric surgery has proven to be an effective method for sustainable weight loss [23]. Many of our participants agreed to the operation as they could not bear with their difficult lives anymore. What further boosted their courage to go through the surgery was strong determination for a meaningful life and the support of their families and friends.

The synthesis of the findings of this study is guided by humanistic theory using Maslow’s hierarchy of needs (Figure 1). This five-stage modal framework helps us explore the challenges faced by people with morbid obesity before surgery and the impacts of weight loss surgery in various domains of their life. The ultimate dream of every human suffering from morbid obesity is to attain desired sustainable weight loss and amelioration of its most associated negative influence. After achieving their fundamental needs, as per Maslow’s hierarchy of needs, they constantly progress in their life to reach the peak towards succeeding in their life and attaining self-actualisation.

In this study, 20 participants who were previously suffering from health-related issues reported to have significant improvement in the control of their disease. The surgery had created a path for them to attain the second stage of the hierarchy, which is to have good physical health. This study confirmed that bariatric surgery helps improve certain diseases such as OSA, diabetes, hypertension, and hyperlipidaemia. This is in line with another study, which demonstrated that large intentional weight loss after surgery in the severely obese subjects cause marked reductions in 2-year incidence of hypertension, diabetes, and hyperlipidaemia [12]. For those suffering from OSA, bariatric surgery helped resolve or at least brought vast improvement for most of the patients [24]. In addition, many participants reported improvement in joint pain with lighter weight. Our finding is also consistent with a previous study, which indicated that weight loss following bariatric surgery is effective at relieving or improving joint pain in weight-bearing joints, such as hip and knee joints with osteoarthritis [25,26]. Many participants were grateful that they had opted for surgery, since the achieved weight loss had brought meaningful changes in their lives, making them free from all debilitating diseases.

Spirituality is the next important aspect in the second stage of the hierarchy. It is an important dimension of quality of life for a human to succeed in their life. Many had experienced positive changes in their spiritual health following weight loss after surgery. For a Muslim, physical fitness is crucial in performing a prayer, reciting the Quran, performing Hajj, and fasting. Excessive abdominal body fat makes the act of praying difficult due to certain movement limitations. Reciting the Quran becomes more challenging as it requires one to control and manipulate their breath, leaving them breathless even to complete one surah. After the surgery, we found that many participants increased the number of their spiritual activities, which included performing more night prayers, going to the mosque, and reciting the Quran. This is in agreement with a previous study in which the surgery helped restore the physical function of a person by extreme weight reduction [26]. Spiritual health is an important aspect of human well-being, as it enables a person to cope with personal existential crises in various aspects of human life: stressful situations, illness, and even the presence of death [27]. Thus, being able to perform more of these activities brings peace and satisfaction to the patients.

The quality of sexual life of an obese person is often impaired. According to the third stage of the hierarchy, the belongingness and the love needs of humans have to be fulfilled for them to progress [15]. In this study, patients shared they experienced sexual difficulties attributed to their high body weight. Such impairment includes difficulties in sexual performance and lack of sexual desire. This finding is in tandem with a previous study conducted among 500 weight loss participants in North Carolina, which reported lack of sexual enjoyment, lack of sexual desire, difficulty with sexual performance, and even avoidance of sexual encounters among their participants, all attributable to excessive weight [28]. This study revealed that those with higher BMI suffer greater impairments in sexual quality of life, and it is more common among obese women than obese men [28]. Bariatric surgery helped these participants as well as our participants to have a tremendous improvement in their sexual life, enabling them to experience pleasant sexual intimacy. This is consistent with a prior study, in which individuals reported an increase in libido, satisfaction, and intercourse frequency [29,30].

It has become a norm that society today devalues overweight and obese individuals and idealises ultra-thinness. This belief system adopted that thinness represents a marker for success in our society and that obesity is to be avoided and feared both because of its impact on the attractiveness and the characterological flaws associated with it, adding to the participant’s low self-esteem and the feeling of isolation. After the surgery, with the great weight loss, the participants in this study managed to make a valuable transition into society. Their steps to the third stage of the hierarchy were meaningful for them, as they no longer faced discriminating gestures in public places, making them feel invited to be around people. This finding agrees with a previous study which was conducted among Brazilian women who had undergone bariatric surgery, indicating that social acceptance was among the significant gains that the participants had experienced [10].

According to Maslow, the most important need for us humans is respect, which comes before true self-esteem or dignity [15]. Similarly in our studies, all the positive changes above stand as a backbone for the participants to move forward to stage four of the hierarchy, which is to be confident with themselves. Their new outlook had boosted their level of self-confidence. With the newfound self-confidence, many of our participants started to involve themselves in activities they found restricted before. They started to enjoy shopping, tried more fashionable clothing, and attended social functions, as their physical appearance was no longer perceived to be bizarre [10]. These positive changes in their social life with weight loss reflected their improved self-esteem after bariatric surgery [11].

Obesity is often associated with sick leave, disability, and workplace injuries. In this study, the patients reported that their obesity had affected their quality of work. Following surgery, they were more enthusiastic about their work. They were finally able to reach the top of the hierarchy. According to Maslow, self-actualisation need is about realising one’s potential and seeking personal growth in their desired field of interest, including creative activities [15]. As for the participants in our studies, they joined courses to expand their knowledge and understanding about their job scope, while others were more satisfied with their work quality. These findings are in line with a previous study that reported a significant improvement in work productivity seen among post-bariatric patients [31].

We also found that many participants were also excited with their new experiences, exploring new hobbies and fun activities either alone or with their families. This is in line with a prior study, which reported that there is a positive change in leisure activity with weight loss and change in sports relationship activity with cardiorespiratory fitness attained following the surgery [32]. These new choices they made helped make their life more colourful than before.

Apart from all these positive changes, some participants faced the complications of the surgery. Half of our participants suffered from GERD, while a few had hair loss and brittle nails. This is not surprising, since a previous study had reported that these complications were common after bariatric surgery [33,34,35]. The type of bariatric surgery seemed to be associated with these complications, where LSG was reported to be more prone to cause GERD compared to LRYGB [33]. Loss of micronutrients may be associated with the hair loss and brittle nails. Although bariatric surgery helped with losing excessive weight, the maintenance of lost weight still depends on healthy lifestyle following bariatric surgery. In this study, three of our participants expressed frustration due to weight gain three years after the surgery due to poor food choices. This was also shown in another study, that patients can still gain weight after the surgery if they fail to adapt to a healthy lifestyle [36,37]. Hence, despite the positive results of bariatric surgery, self-discipline and adoption of a healthier lifestyle are paramount in those undergoing bariatric surgery to maintain their lower BMI.

The power of this study is within the phenomenological analysis involving face-to-face in-depth interviews, gaining profound insight into analysing changes in life that the participants experience before and after the weight loss surgery. The data that divulged into physical and emotional changes faced by these people suffering from morbid obesity from this qualitative model in Kelantan are valued to expedite the assessment and management in a clinical setting. Understanding the benefits and complications of the surgery can assist in the supportive role we as a community need to comprehend for us to bring this surgery to the eyes of the public.

However, this study is limited by the small sample size, and confined to Malay ethnicity. A different ethnicity may have different cultural beliefs that influence their experience. Moreover, this study was conducted in a tertiary hospital where none of the participants experience discrimination. Differing themes may be observed from post-bariatric patients from different ethnicities, religions, age groups, and sociodemographic profiles, which require further study.

## 5. Conclusions

Our findings suggest that all the participants experienced many positive changes in their lives after the surgery. Bariatric surgery has given them a second chance in life. Benefits were attained and seen in multiple aspects of life, starting from disease resolution and improvement in disease control, increased spirituality, improved self-esteem, sexual health, and productivity. Despite some suffering from side effects after the surgery, those effects seem minute in their eyes compared to the value changes they have gained. The benefits of bariatric surgery should be brought to the public with the help of collaborative efforts from both the stakeholders and healthcare practitioners, so that more people will benefit from it. Although bariatric surgery had proved to be the answer to most of the problems related to morbid obesity, it is crucial for the person involved to learn to adapt to a healthy lifestyle and practice mindful eating habits following the surgery to achieve and maintain an ideal weight.

## Figures and Tables

**Figure 1 ijerph-19-06009-f001:**
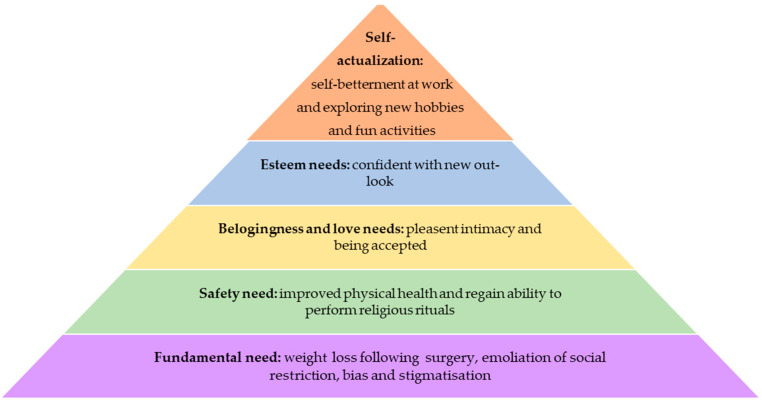
Adapted theoretical framework from Maslow’s hierarchy of needs.

**Table 1 ijerph-19-06009-t001:** Characteristics of the morbidly obese people with bariatric surgery (*n* = 21).

ID	Age	Gender	Comorbidities	Year of Operation	Type of Operation	Pre-Operation BMI (kg/m^2^)	Post-Operation BMI (kg/m^2^)
ID 1	36	F	HPT, DM	2019	SG	53	41.8
ID 2	60	F	HPT	2017	MGB	49.3	35.4
ID 3	45	F	Knee OA, OSA	2019	RYGB	37.2	27.3
ID 4	45	F	ASTHMA	2019	SG	70.1	50.6
ID 5	39	M	HPT	2020	SG	67.2	55.7
ID 6	34	F	HPT	2017	SG	45.8	32.5
ID 7	41	M	DM, HPT	2018	SG	52	41.6
ID 8	59	M	Knee OA	2017	SG	41.9	29.0
ID 9	43	F	Nil	2019	SG	42.5	30.1
ID 10	47	F	DM, HPT, OSA	2019	SG	50.9	36.6
ID 11	43	F	HPT, OSA	2019	SG	57	45.0
ID 12	61	F	HPT	2019	RYGB	42.3	30.0
ID 13	35	F	HPT, DM	2017	RYGB	40.3	28.9
ID 14	29	M	OSA	2018	SG	45.8	36.3
ID 15	40	M	HPT	2018	SG	57.7	40.0
ID 16	35	M	OSA, IFG, Cystic acne	2018	SG	41.8	31.9
ID 17	42	F	HPT, OSA	2020	SG	54.5	40.3
ID 18	41	F	Knee OA	2018	SG	65.6	48.5
ID 19	28	F	Acne vulgaris	2017	SG	91.4	72.5
ID 20	47	F	HPT, OSA	2019	RYGB	37.2	27.7
ID 21	45	F	DM, HPT, HPL	2018	RYGB	37.1	28.7

DM, diabetes mellitus; HPT, hypertension; HPL, hyperlipidaemia; ID, identification; IFG, impaired fasting glucose; MGB, mini gastric bypass; OA, osteoarthritis; OSA, obstructive sleep apnoea; SG, sleeve gastrectomy; RYGB, Roux-en-Y gastric bypass.

**Table 2 ijerph-19-06009-t002:** Life experiences of the morbidly obese person with bariatric surgery (*n* = 21).

Themes	Subtheme	Axial Coding
**Social restraint**	Burdensome to carry out activities of daily living	Tired to go shopping and other activities in life
	Difficult to perform the spiritual act	Feel embarrassed for occupying a lot of space in the prayer line Breathless to walk to the mosque
	Struggle at workplace	Difficult to climb up to the classes at a higher level at school Exhausted to walk around the class to monitor the students
	Unpleasant travelling	Lack of attractive attire options Inadequate public facilities Embarrassing life event
**Experiencing bias and stigmatisation**	Body-shaming from their family members	Labelling linked to an animal
	Discrimination	Discriminating gestures by the surrounding people in public places Judgmental remarks on work quality by bosses
**Bringing new life**	Improved physical health	Resolution of diabetes, hypertension and high cholesterol OSA symptoms subsided Disappearance of skin pigmentation and acne Vanishing debilitating knee pain
	Regain the ability to perform religious rituals	Reciting Al-Quran at ease Able to perform spiritual acts unhindered
	Pleasant intimacy	Increased sexual drive Can have sex longer Improved couple satisfaction
	Exploring new hobbies and fun activities	Confident to travel alone Fun long drive with family Sports as new hobbies Enjoying sandy beach and exciting games at a theme park
**Boosting self-esteem**	Being accepted	Felt valued by family and friends No more discrimination
	Confident with new outlook	Increased self-esteem Excited wearing trendy clothing
	Self-betterment at work	Improved working quality Enthusiastic about expending knowledge on new work scope
**Facing the negative side of surgery**	Facing side effects after surgery	Suffering from GERD Getting bald after surgery
	Regain weight	Failure to be mindful of diet choices

## Data Availability

Data are contained within the article.

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
