# Peer review of "Lived Experience after Bariatric Surgery among Patients with Morbid Obesity in East Coast Peninsular Malaysia: A Qualitative Study"

_ijerph, 2022, doi:10.3390/ijerph19106009_

Round 1

Reviewer 1 Report

This is an interesting but lengthy article about the experiences of 21 adults undergoing bariatric surgery in Malaysia. Many of the references are quite old and the article could be shortened a bit without losing valuable information.

Reviewer 2 Report

Introduction

This is a very interesting study with a large number of patients for this type of analysis.  I just have a few comments.  The exploration of spirituality is particularly important.  However, the authors need to make clearer what is being attributed to patients and not editorialize in the results section. 

Here are some specific concerns:

The data shows that Asian populations have a lower mean BMI than western populations, but there is less certainty with adiposity.  I Would consider changing “Asian populations are lean when compared to western populations…” to “Asian populations, in general, have a lower BMI compared to western populations for similar levels of adiposity.”

I feel there is editorializing in the results in general.   I suggest making clearer what is attributed to the patients throughout this section.  More of the text items should be in quotes rather than being attributed to the authors. 

For example:

In 3.5.3 “Obesity affects sexual health issues more than we usually think” should be trimmed to “Obesity affects sexual health.”   I would drop “unattractive look.”

Change, ” unattractive appearance” to “what patients perceive as unattractive appearance..” or put it in quotes.

With “colossal statue” either put it in quotes or change it to “large.”

In general, change to person-first terminology.  “Overweight and obese people” should be changed to “people who are overweight or obese.”

“Those side effects left no regret in them as they treasure the life they are living now” this should be changed or put in quotes. 

Did the patients say they have no regrets about the surgery or is this the author’s opinion?

The Discussion:

There are several statements made that go beyond the data presented.

I do not think the authors can make the statement,” All the suffering had ended after significant weight loss attained with the help of surgery. Patients had a chance to enjoy a brand-new life as they dream about.”  This is beyond the data presented in the results. 

The authors do not present any descriptive statistics, so no conclusions can be drawn about surgical results other than what the patients perceive.   This needs to be made clearer and statements like “In this study, all the patients who were previously suffering from health-related is-594 sues experienced complete resolution or at least significant improvement in the control of 595 their disease,” needs to be dropped. 

Reviewer 3 Report

title: indicate this is a qualitative study 

Abstract: 

line 18 a phenomenological approach and in-depth interviews, please clarify this is one or two methods you are using in your study?

the description of the participants, the main findings should be added in the results part of the abstract

Introduction

line 38-43 this paragraph is unnecessary, using the obesity related health consequences instead.

Methods

This part should be rewritten

line 103-111 the introduction this method, its advantage and disadvantage can be introduced and discussed in the discussion part.

the inclusion and exclusion criteria should be described

the study site i.e. where the interviews were conducted should be described in detail as this would affect the results

Results

the results part should be re-organized 

Table 1 need to be revised a summarized result should be provided, such as the how many male, and female participants,  the time period after operation, etc..

Conclusion 

the conclusion should be made cautiously, it is a kind of option for obesity treatment, but not the sole one. The change of behavior should be emphersized as well.  

Round 2

Reviewer 2 Report

The authors have done an excellent job of addressing all the reviewer comments.  I just have one more very minor suggested edit at this time:   

On page 5, I would change:

"Hence, despite the wonderful results of bariatric surgery self-discipline and adoption of healthier lifestyle is paramount in those undergoing bariatric surgery are to maintain their lower BMI."

to 

"Hence, despite the positive results of bariatric surgery, self-discipline and adoption of a healthier lifestyle are paramount in those undergoing bariatric surgery to maintain their lower BMI."
